# Beavers, Bugs and Chemistry: A Mammalian Herbivore Changes Chemistry Composition and Arthropod Communities in Foundation Tree Species

Rachel M. Durben [1,2], Faith M. Walker [1,2,3], Liza Holeski [1,2,4], Arthur R. Keith [1,2], Zsuzsi Kovacs [1,2], Sarah R. Hurteau [5], Richard L. Lindroth [4] , Stephen M. Shuster [1,2] and Thomas G. Whitham [1,2,*]

[1] The Environmental Genetics and Genomics Laboratory, Department of Biological Sciences, Northern Arizona University, P.O. Box 5640, Flagstaff, AZ 86011, USA; rdurben@gmail.com (R.M.D.); faith.walker@nau.edu (F.M.W.); Liza.Holeski@nau.edu (L.H.); arthur.keith@nau.edu (A.R.K.); sik5@nau.edu (Z.K.); Stephen.Shuster@nau.edu (S.M.S.)
[2] The Center for Adaptable Western Landscapes, Department of Biological Sciences, Northern Arizona University, P.O. Box 5640, Flagstaff, AZ 86011, USA
[3] School of Forestry, Northern Arizona University, Flagstaff, AZ 86001, USA
[4] Department of Entomology, University of Wisconsin-Madison, Madison, WI 53706, USA; richard.lindroth@wisc.edu
[5] Geography and Environmental Studies Department, University of New Mexico, Albuquerque, NM 87131, USA; sarah.hurteau@tnc.org
[*] Correspondence: Thomas.Whitham@nau.edu

**Abstract:** The North American beaver (*Castor canadensis* Kuhl) and cottonwoods (*Populus* spp.) are foundation species, the interactions of which define a much larger community and affect a threatened riparian habitat type. Few studies have tested the effect of these interactions on plant chemistry and a diverse arthropod community. We experimentally examined the impact of beaver foraging on riparian communities by first investigating beaver food preferences for one cottonwood species, Fremont cottonwood (*P. fremontii* S. Watson), compared to other locally available woody species. We next examined the impact of beaver foraging on twig chemistry and arthropod communities in paired samples of felled and unfelled cottonwood species in northern Arizona (*P. fremontii*) and southwestern Colorado (narrowleaf cottonwood, *P. angustifolia* James, and Eastern cottonwood, *P. deltoides* W. Bartram ex Marshall). Four major patterns emerged: (1) In a cafeteria experiment, beavers chose *P. fremontii* six times more often than other woody native and exotic species. (2) With two cottonwood species, we found that the nitrogen and salicortin concentrations were up to 45% greater and lignin concentration 14% lower in the juvenile resprout growth of felled trees than the juvenile growth on unfelled trees (six of seven analyses were significant for *P. fremontii* and four of six were significant for *P. angustifolia*). (3) With two cottonwood species, arthropod community composition on juvenile branches differed significantly between felled and unfelled trees, with up to 38% greater species richness, 114% greater relative abundance and 1282% greater species diversity on felled trees (six of seven analyses with *P. fremontii* and four of six analyses with *P. angustifolia* were significant). The above findings indicate that the highest arthropod diversity is achieved in the heterogenous stands of mixed felled and unfelled trees than in stands of cottonwoods, where beavers are not present. These results also indicate that beaver herbivory changes the chemical composition in 10 out of 13 chemical traits in the juvenile growth of two of the three cottonwood species to potentially allow better defense against future beaver herbivory. (4) With *P. deltoides*, only one of five analyses in chemistry was significant, and none of the four arthropod community analyses were significant, suggesting that this species and its arthropod community responds differently to beaver. Potential reasons for these differences are unknown. Overall, our findings suggest that in addition to their impact on riparian vegetation, other mammals, birds, and aquatic organisms, beavers also may define the arthropod communities of two of three foundation tree species in these riparian ecosystems.

**Keywords:** beaver; *Castor canadensis*; cottonwood; *Populus* spp.; chemistry; arthropod communities; foundation species

## 1. Introduction

The ecological processes that structure communities are fundamental aspects of ecology and evolution. Holling [1] proposed that "a small set of plant, animal, and abiotic processes structure ecosystems across scales in time and space". In other words, not all species are equal, and a few are likely to contribute disproportionately to the structure and evolution of communities and ecosystems. Species that are strong interactors, affecting many other species and modifying their environments (e.g., dam building and selective foraging by beavers; shading and cooling of the understory by tree canopies) likely play a major role in structuring communities. Dayton [2] defined a foundation species as "a single species that defines much of the structure of a community by creating locally stable conditions for other species, and by modulating and stabilizing fundamental ecosystem processes". Ellison et al. [3] reviewed this literature and concluded that dominant species, keystone species, ecosystem engineers and other strong interactors all fall within this category of foundation species. Because all ecosystems of the world likely have multiple and potentially interacting foundation species, it is especially important to identify and understand the interactions of these species as their loss due to global change (including climate change, invasive species, and altered species interactions) could cascade to affect the rest of the community [4]. Keith et al. [5,6] proposed and experimentally tested the interacting foundation species hypothesis and found that the interactions of two foundation species better explained community diversity, stability and species interaction networks than either one alone.

Beavers and cottonwoods are considered foundation species because their presence in an ecosystem has far-reaching effects on community dynamics and ecosystem processes [3]. By felling trees that are used for food and construction activities (e.g., building dams that slow water flow and create ponds and lodges), beavers affect fish populations [7], stream flow and water temperature [8,9], nutrient cycling and availability [9–11], individual species and communities of arboreal arthropods [12–14], the composition of mixed cottonwood riparian forests and their genetic composition [15], communities of aquatic arthropods [16], vegetation diversity [17], tree architecture [18], and genetic diversity and productivity at the individual tree level [19]. Even in areas where beavers are no longer active but have cut trees and built dams in the recent past, their impact is evident in the structure of the riparian zone and has long-term effects on vegetation types [9,11,20–23]. Compared with the amount of land they cover, riparian corridors contain disproportionately high biodiversity [24] and are an important resource for birds and mammals. In the arid southwest USA, the narrow riparian "ribbon of green" (Figure 1A) has been shown to support exceptionally high bird [25] and vegetation diversity [26]. However, the extent to which beavers interact with a foundation tree species to affect arthropod diversity in these southwestern stream systems has not been studied (but see [14]).

One mechanism by which beaver herbivory might affect arthropod diversity is by altering the chemistry of the plants they browse. To determine whether there was a difference in twig chemistry between the juvenile resprout growth of beaver-felled trees (Figure 1B), and the juvenile shoots at the base of unfelled trees, we collected samples from each to assess the concentrations of a suite of chemicals known to be present in these tissues. Since beavers eat the inner bark, or cambium layer of trees [27], we chose to sample twig tissue to quantify how beaver herbivory might affect the phytochemistry of resprouting cottonwoods for future feeding by beavers. While we cannot extrapolate on how twig chemistry may influence foliar arthropods, we were also interested in understanding how herbivory by beavers may differentially affect arthropods on multiple cottonwood species, which has not previously been examined in this system.

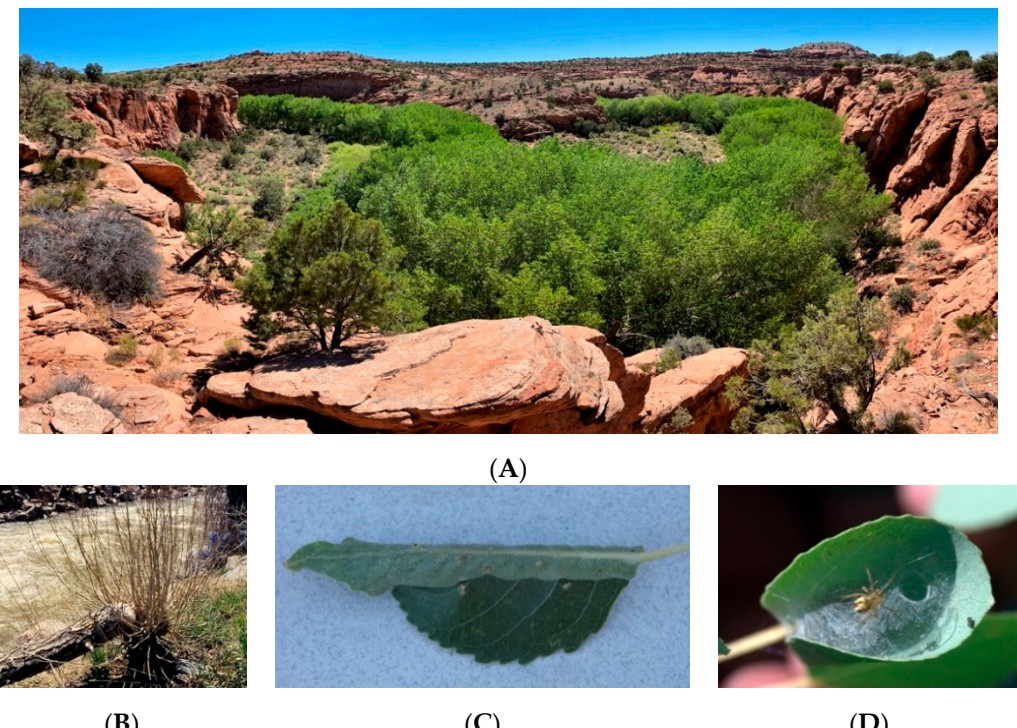

**Figure 1.** (**A**) Riparian "ribbon of green" of *P. fremontii* near Moab, UT; (**B**) Resprout growth of *P. angustifolia* after felling by beavers; (**C**) Leaf rolling moth (*Anacampsis niveopulvella* Chambers, 1875; an indicator species of resprout growth after *P. fremontii* felling by beavers); (**D**) Leaf rolling moth larvae, a member of the arthropod community influenced by beaver herbivory, in the family Tortricidae (also indicator species of resprout growth) create habitat for other arthropods, such as this spider.

To examine how beaver herbivory on Fremont cottonwood (*P. fremontii*), Eastern cottonwood (*P. deltoides*) and narrowleaf cottonwood (*P. angustifolia*) in different southwestern river systems may change twig chemistry and arthropod communities (see representative arthropod species Figure 1C,D), we tested three hypotheses: (1) Beavers show a preference for Fremont cottonwood over three other native species and one invasive woody species common to the area. (2) The resprouted juvenile stems of unfelled trees and trees felled by beavers differ in their levels of nitrogen, carbon, lignin, condensed tannins and the phenolic glycosides salicortin and HCH-salicortin. (3) The arthropod communities in paired samples of juvenile foliage on felled and unfelled trees differ in their multivariate community parameters including metrics such as species richness and abundance.

Testing these hypotheses will increase our knowledge of how the interactions of plant and mammalian foundation species impact a diverse arthropod community. While several studies have examined the impacts of beavers on aquatic arthropods [16] and single arboreal arthropod species [12,13], only one other study that we are aware of examined the impacts of beaver felling of trees on a diverse arboreal arthropod community on narrowleaf cottonwood in another region [14]. Understanding the interactions among ecosystem-defining species (beavers, different species of cottonwoods and other native and exotic trees/shrubs), phytochemistry, and arthropod communities will allow for more holistic approaches to management and conservation in these fragile southwest riparian habitats.

## 2. Materials and Methods

### 2.1. Study Sites

Arizona Sites—We examined the effects of beaver herbivory on the arthropod communities and chemistry composition of Fremont cottonwood by studying the riparian zone along two streams in Arizona. The Verde River (altitude 976 m) is a perennial stream that

begins near Paulden, Arizona and terminates at its confluence with the Salt River east of Phoenix, Arizona. Dry Beaver Creek (altitude 987 m) is an intermittent tributary north of the Verde River and experiences beaver herbivory year-round as conditions allow (during spring high flows, through the monsoon season and during wet winter months). The site surveyed is also adjacent to several permanent pools, which beavers can access even when the channel is dry. All studies were conducted at the same locations on each stream.

Colorado Sites—To examine beaver interactions with other cottonwood species, we investigated sites in southwestern Colorado on the San Miguel River to test our hypotheses on two additional abundant *Populus* species—Eastern and narrowleaf cottonwood. This river encompasses a naturally occurring hybrid zone of trees with narrowleaf at its highest altitudes near Telluride, Colorado (altitude 2667 m) and Eastern cottonwood at the lower altitudes near Naturita, Colorado (altitude 1651 m). This hybrid zone is formed by the bidirectional introgression of both species at intermediate elevations supporting populations of $F_1$ hybrids and backcrosses of both narrowleaf and Eastern cottonwood [28]. The narrowleaf site is located just upstream of the confluence of Leopard Creek and the San Miguel River. This site supports a highly active beaver population with a complex of dams and lodges. Incorporating the results of this population allowed for a broader geographic interpretation of patterns regarding chemistry composition and arthropod community diversity across three related tree species.

*2.2. Cafeteria Study*

In an earlier study to determine which species of local woody vegetation beavers prefer, we conducted two cafeteria trials at each of the two beaver ponds at Dry Beaver Creek between 27 October and 10 November 2004. Each trial involved 25 food samples per pond (100 total samples), consisting of five food samples each of four abundant native species growing within 30 m of the stream (Fremont cottonwood, *Populus fremontii*; Arizona sycamore, *Platanus wrightii* S. Watson; velvet ash, *Fraxinus velutina* Torr; coyote willow, *Salix exigua* Nutt.) and one exotic highly invasive species (tamarisk, *Tamarix ramosissima* Ledeb.). Each food sample was a 2–3 cm diameter branch haphazardly collected from along the creek from unfelled trees. Because beavers may forage within a 30 m radius of their pond, all cafeteria food samples were collected within this distance from each pond [18]. Note that this cafeteria experiment was only performed on Fremont cottonwood in Arizona and not on the other two cottonwood species in southwest Colorado.

We labeled and flagged each food sample with a number and a letter to denote the species and haphazardly placed them along each pond's edge near canopy cover to minimize exposure of beavers to predators. We recovered trial 1 after three days, removing all uneaten food samples prior to trial 2. Food samples for trial 2 were removed after 10 days. Removal for each trial involved searching ponds and dams for remaining branches. Although there was direct evidence of cottonwood being eaten, if other samples were missing, they were considered as eaten.

*2.3. Chemistry Collection and Analyses*

To analyze the chemical profiles of the three cottonwood species, we collected 45 cm of terminal branch growth from the juvenile zone of each tree. At all Arizona sites, twigs were collected from 24 pairs of trees at each site in November of 2008. Twigs from 18 pairs of Eastern cottonwood and 22 pairs of narrowleaf cottonwood were collected on the San Miguel River in October of 2009. We selected trees as pairs, where trees within a pair are of similar basal trunk diameter to ensure that each group contains trees of the same age class and of similar distance from the water to ensure similar resource availability. One tree in each pair had not been felled by beaver and served as a control. The second tree in each pair had been previously felled by a beaver and had resprouted from the trunk. The same trees selected for chemistry collection were also surveyed for arthropod communities. While Eastern and Fremont cottonwood are not clonal, narrowleaf can be very clonal [29]. To ensure we were not collecting samples from narrowleaf clones, we sampled from widely

separated trees. Walker et al. [unpublished] used microsatellite markers to genotype the same trees used in this study and found only one instance of clonality in narrowleaf cottonwood on the San Miguel. While chemistry can vary annually, Cole et al. [30] found in aspen (*P. tremuloides* Michx.) that variation is minimal in contrast to strong genetic and ontogenetic variation in chemistry.

Note that number discrepancies are due to trees being felled by beavers between sampling periods resulting in their lacking sufficient resprout growth for subsequent sampling. Due to permitting issues, we were only able to cage unfelled trees to prevent beaver herbivory at the Verde River. We originally had 24 pairs marked at Dry Beaver Creek and 30 at the Verde River. We collected chemistry samples from only 24 pairs at the Verde River to have equal numbers of samples at each site, due to a smaller area of beaver activity at Dry Beaver Creek. Since we collected chemistry samples before our first arthropod surveys in Arizona, we lost 5 unfelled trees to beaver herbivory at Dry Beaver Creek between sampling for chemistry and conducting arthropod surveys. In Colorado, we collected twig samples after conducting arthropod surveys (on 25 pairs of trees at each site), and lost 3 unfelled narrowleaf and 7 unfelled Eastern cottonwood to beaver herbivory between sampling periods.

Collections were made in accordance with the methods of Rehill et al. [31]. The twigs were flash frozen on dry ice in the field, then freeze dried, ground in a Wiley mill to pass a 20-mesh (lignin analysis) or 40-mesh (all other analyses) screen and stored at −20 °C at Northern Arizona University. All samples were then sent to the University of Wisconsin-Madison for chemical analysis of nitrogen, carbon, lignin, condensed tannins, and a suite of phenolic glycosides according to the methods of Lindroth et al. [32].

Nitrogen and carbon were quantified using a Thermo Finnigan Flash 1112 elemental analyzer (Thermo Finnigan, San Jose, CA, USA). Lignin was quantified as described in [33], using an Ankom 200 Fiber Analyzer (ANKOM Technology, Macedon, NY, USA). Condensed tannins and phenolic glycosides are the main secondary metabolites in *Populus* [34–37]. To assess condensed tannin content, we used the acid butanol assay [38] with purified narrowleaf cottonwood condensed tannins as standards, as described by Rehill et al. [37]. We assessed phenolic glycosides (salicortin, salicin, and HCH-salicortin) using high performance thin-layer chromatography (HPTLC) as reported by Lindroth et al. [32]. Salicin standard was obtained from Sigma-Aldrich, while salicortin and HCH-salicortin were purified from Fremont and narrowleaf tissue. We report the results of each chemical assay as percent (%) dry weight.

### 2.4. Arthropod Surveys

To test the hypothesis that beaver herbivory alters arthropod communities on Fremont, Eastern and narrowleaf cottonwoods, we surveyed arthropod communities on trees at Dry Beaver Creek and the Verde River in May and August 2009, and at the San Miguel River in Colorado in May/June of 2009. We selected trees as pairs, where trees within a pair are of similar basal trunk diameter, to ensure that each group contains trees of the same age class and of similar distance from the water to ensure similar resource availability. One tree in each pair had not been felled by beaver, and served as a control. The second tree in each pair had been previously felled by a beaver and had resprouted from the trunk. We conducted non-destructive, 20-min visual surveys on 19 tree pairs at Dry Beaver Creek, 30 tree pairs at the Verde River and 50 tree pairs on the San Miguel River, for a total of 198 trees: 98 Fremont, 50 narrowleaf and 50 Eastern cottonwoods. On previously felled trees, we conducted arthropod surveys on resprout growth, while on the unfelled trees, we surveyed branches from the juvenile section (below 2 m). To survey the same amount of biomass per tree, following the methods of Wimp et al. [39,40], we measured each branch until we reached a total of 20 mm of branch diameter for surveying to standardize for the relatively small amount of growth found on resprout trees. We then observed these branches for 20 min and recorded all arthropods present on the trees to the finest taxonomic level possible. We collected and later identified those specimens that were not identifiable

in the field. The arthropod specimens collected are available in a reference collection in the Arthropod Museum at Northern Arizona University.

*2.5. Statistical Analyses*

To determine the foraging preferences of beavers, we compared the trials for each pond and the number of branches of each of the five tree types eaten using R × C *G*-tests. To determine whether chemistry profiles and arthropod communities differed between felled and unfelled trees, we used Multi-Response Permutation Procedure (MRPP) to generate Non-metric Multidimensional Scaling scores using PC-ORD version 5 [41]. For community analysis, we relativized abundance data for arthropod matrices, using species maximum because extremely common species can drive abundance patterns, [42,43]. MRPP calculates within-group distance—δ, where small values indicate similar groups and large values indicate more difference within groups. These relationships are represented by an A-value between 1 and −1, where values greater than zero but less than 1 signify identical within-group similarity but differences between groups; zero represents completely random grouping; and values less than zero indicate more within-group differences than would be expected by chance. Non-metric Multidimensional Scaling is then used to visualize the data generated by the MRPP dissimilarity matrix. After the matrix has been constructed, NMDS searches for the configuration with the least stress, using multiple iterations of an algorithm. The solution with the lowest stress is then selected and a graphical ordination is created. NMDS ordinations are recognized as the best method for visually representing similarities or dissimilarities in multivariate data such as our chemistry or community data, largely because there are no assumptions of normally distributed data, an assumption that is often extremely difficult to satisfy [44]. Indicator species analyses use the methods of Dufrene and Legendre [45] to produce indicator values for each species in each of a number of groups.

Samples for chemistry analyses were collected from Arizona in November 2008, and from Colorado in October 2009. Arthropod surveys were conducted on two different occasions in Arizona and on one occasion in Colorado (AZ: May 2009, August 2009; CO: May/June 2009). Because the Arizona and Colorado sampling sites were widely geographically separated, and supported different cottonwood species, we analyzed the sites from each state separately. Additionally, because samples collected within each state were distinct in when they were collected as well as heterogeneous in the arthropod species and abundances they included (see Results), we considered each set of samples within each location as a separate analysis. For these analyses, we included Site, Treatment and Site × Treatment as factors in our MRPP analyses of chemistry and arthropod community analyses, as described above. When multivariate analyses showed significant interactions between Site and Treatment effects, we analyzed each site separately, and if the subdivided data were significant, we compared individual results for each of the five phytochemicals (carbon, nitrogen, lignin, salicortin, HCH-salicortin) using pairwise tests. In all cases in which analyses were subdivided, we Bonferroni-adjusted our criteria for significance by the number of tests performed. To examine chemical differences as well as total arthropod abundance and species richness in beaver-felled and unfelled trees, we conducted univariate analyses using the statistical software package JMP in 8.0.1 [46]. Data were first checked for normality using the Shapiro–Wilk goodness of fit test. If normality assumptions were met, the data were analyzed using a paired *t*-test. For abundance data, the data were relativized by species maximum to correct for highly abundant species that might drive the distribution pattern. This was carried out by dividing the total abundance value for each species in a treatment by the largest observed value, changing the scale of the data to values between 1 and zero. If normality assumptions were then met, data were analyzed with a paired *t*-test. For all other measurements, or if normality assumptions were still violated with abundance data, we used a Wilcoxon signed-rank test.

## 3. Results

### 3.1. Beaver Cafeteria Feeding Experiment

In trial 1 (collected after three days), 80% of cottonwood branches were eaten but no branches were eaten of tamarisk, sycamore, ash or willow. In trial 2 (collected after 10 days), 100% of cottonwood branches were eaten, 10% (1/10) and 20% (2/10) of the branches of ash and willow were eaten, respectively, and no branches of tamarisk and sycamore were eaten. To meet R × C *G*-test assumptions [47], we collapsed the results of the trials into a matrix with two rows (eaten; not eaten) and two columns (cottonwood; non-cottonwood). The resulting 2 × 2 *G*-test showed a significant preference by beavers for cottonwood branches over the other available woody vegetation (*G* = 64.2, df = 1, *p* < 0.0001, *N* = 100; Figure 2).

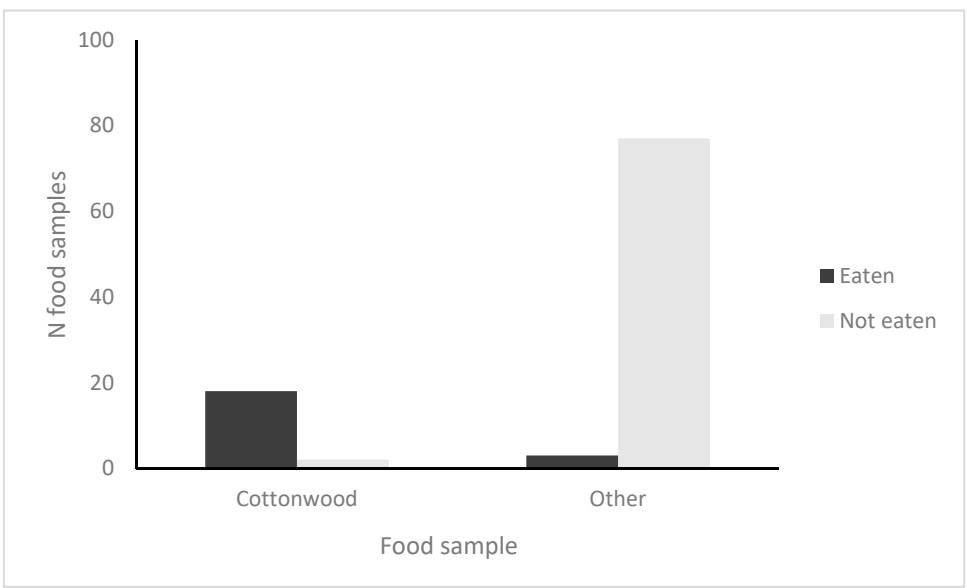

**Figure 2.** Results of foraging preferences for beavers at Dry Beaver Creek, AZ in four trials of 25 (100 total samples). Beavers ate 18/20 (90%) of cottonwood samples and 3/80 (4%) of non-cottonwood samples; the data collapsed into a 2 × 2 table for analysis were significant (*G* = 64.2, df = 1, *p* < 0.0001).

### 3.2. Tree Phytochemical Responses to Beaver Herbivory

Arizona Sites: Our MRPP/NMDS analyses of Fremont cottonwood trees indicated that total twig chemistry differed between study locations as well as between the juvenile resprout growth of felled and the juvenile growth of unfelled Fremont cottonwoods. For the Arizona study, with 24 pairs of trees, we found significant effects for Site (Dry Beaver Creek, Verde River; A = 0.125; *p* < 0.0001) and Treatment (Felled, Unfelled; A = 0.016, *p* = 0.0163; Figure 3A) and a significant Site × Treatment interaction (A = 0.144, *p* < 0.001). Separate MRPP/NMDS analysis of twig chemistry for each AZ site (Bonferroni-adjusted α = 0.05/2 = 0.025) showed a significant chemical difference between felled and unfelled trees at Dry Beaver Creek (DBC; A = 0.025, *p* = 0.019), but no significant chemical differences between similar trees at Verde River (VR; A = 0.017, *p* = 0.063). Samples from resprout growth on felled Fremont cottonwood at Arizona sites showed 14% higher nitrogen concentration than those of the juvenile growth of unfelled trees (*p* < 0.0001). Our Bonferroni-adjusted pairwise Wilcoxon signed-rank tests for five phytochemicals (Table 1: carbon, nitrogen, lignin, salicortin and HCH-salicortin; adjusted α = 0.05/(5 + 2) = 0.007) showed significant differences between felled and unfelled trees for nitrogen (*p* = 0.0001), lignin (*p* = 0.0035), salicortin (*p* = 0.002) and HCH-salicortin (*p* = 0.009) at Dry Beaver Creek only. We found no significant differences between felled and unfelled trees for carbon (*p* = 0.10) or for HCH-salicortin at the Verde River (*p* = 0.035).

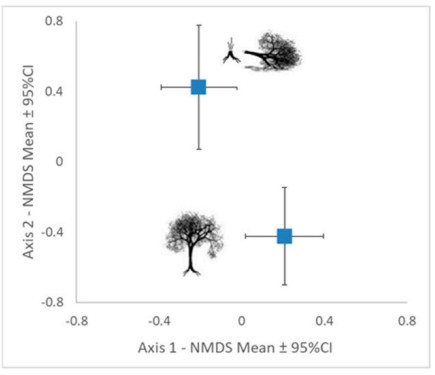

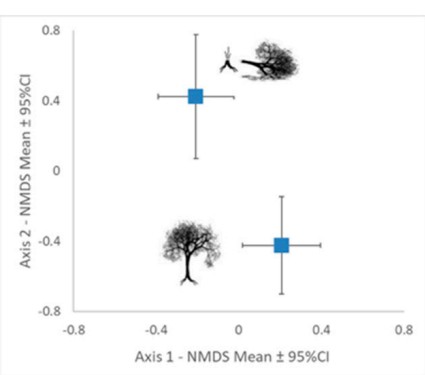

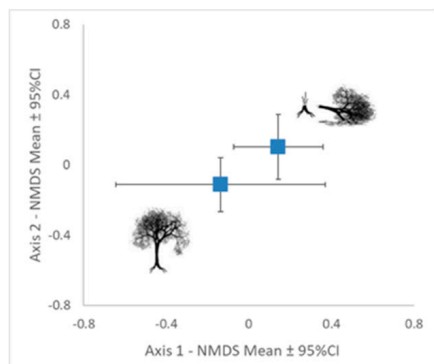

(**A**) Arizona Sites, Fremont　　(**B**) San Miguel, Narrowleaf　　(**C**) San Miguel, Eastern

**Figure 3.** Non-metric multidimensional scaling (NMDS) compared chemistry profiles of juvenile felled and unfelled cottonwood twigs: (**A**) Fremont cottonwoods in Arizona had significant effects for Site (Dry Beaver Creek, Verde River; A = 0.125; $p < 0.0001$) and Treatment (Figure shown: Felled, Unfelled; A = 0.016, $p = 0.0163$) and a significant Site × Treatment interaction (A = 0.144, $p < 0.001$). (**B**) In narrowleaf cottonwoods on the San Miguel River, there was also a significant difference between the chemical composition of resprout growth and twigs from unfelled trees (A = 0.123, $p < 0.0001$); (**C**) Eastern cottonwoods on the San Miguel River, CO showed no significant difference in chemistry profiles between felled and unfelled trees (A = −0.005, $p = 0.59$). Icons represent felled and unfelled profiles; centroids are mean values with 95% confidence intervals.

Colorado Site: Our MRPP/NMDS analysis of narrowleaf and Eastern cottonwood trees on the San Miguel River also supported the hypothesis that twig chemistry differed between the juvenile resprout growth of felled and juvenile growth of unfelled narrowleaf trees (A = 0.123, $p < 0.0001$, Figure 3B), but not for Eastern cottonwood trees (A = −0.005, $p = 0.59$; Figure 3C). Our Bonferroni-adjusted pairwise Wilcoxon signed-rank tests for six phytochemicals (Table 1: carbon, nitrogen, lignin, salicortin, HCH-salicortin and condensed tannins; adjusted $\alpha = 0.05/6 = 0.008$) showed significant differences between felled and unfelled narrowleaf trees for carbon ($p = 0.001$), nitrogen ($p = 0.0001$), lignin ($p = 0.004$) and salicortin ($p = 0.0003$). Contrary to what has been suggested by other research [27], we found no significant differences between felled and unfelled trees for condensed tannins ($p = 0.11$) or for HCH-salicortin ($p = 0.71$). Eastern cottonwood only showed a significant difference between felled and unfelled trees with respect to nitrogen ($p = 0.047$), but this significance did not hold up under Bonferroni adjustments.

### 3.3. Arthropod Responses to Herbivory

Arizona Sites: Our MRPP/NMDS analyses of Fremont cottonwood trees indicated that arthropod communities differ between study locations as well as between the juvenile resprout growth of felled and unfelled Fremont cottonwoods. We found significant differences in the arthropod communities on one collection date (August 2009). We also found significant Site, Treatment and Site x Treatment effects in both sampling dates (Site: AZ May 2009: A = 0.017; $p < 0.0001$; August 2009: A = 0.023; $p < 0.0001$; Treatment: Felled, Unfelled: AZ May 2009: A = 0.013; $p < 0.005$; August 2009: A = 0.012; $p < 0.0001$; Site × Treatment: AZ May 2009: A = 0.022; $p < 0.0001$; August 2009: A = 0.036; $p < 0.0001$). We considered separate MRPP/NMDS analysis of arthropod communities on felled and unfelled trees for each AZ site unjustified, given the unusual rigor of our Bonferroni-adjusted criterion for this comparison ($\alpha = 0.05/6 = 0.008$).

**Table 1.** Chemistry composition was significantly different in twigs collected from juvenile resprout growth than in juvenile twigs from unfelled trees in Arizona and in Colorado. When we analyzed each chemical independently, we found no significant difference between Arizona sites for carbon, lignin or nitrogen, so these sites were pooled for analyses. The directionality of each difference, if any, along with the corresponding *p*-value is reported. All analyses were conducted using a non-parametric Wilcoxon-Signed Rank Test, and Bonferroni adjusted for an α of 0.007 for Fremont and Eastern cottonwood and 0.008 for narrowleaf cottonwood. *p*-values with two asterisks ** are significant with Bonferroni-adjusted α. *p*-values with one asterisk * are significant prior to Bonferroni adjustments. In Fremont cottonwood, we found significant differences in 5 out of 7 analyses when using Bonferroni adjustments and in 6 out of 7 analyses without Bonferroni adjustments. In narrowleaf cottonwood, we found significant differences in 4 out of 6 analyses both with and without Bonferroni adjustments. In Eastern cottonwood, we found a significant difference in 1 out of 5 analyses (nitrogen), but significance was not maintained with Bonferroni adjustments.

| Chemistry | Dry Beaver Creek, AZ Fremont (*N* = 24 Pairs) | Verde River, AZ, Fremont (*N* = 24 Pairs) | San Miguel, CO Narrowleaf (*N* = 22 Pairs) | San Miguel, CO Eastern (*N* = 18 Pairs) |
|---|---|---|---|---|
| **Carbon** | Felled = Unfelled μ (felled) = 46.53 μ (unfelled) = 46.44 SE = 0.19 *p* = 0.10 | | Felled > Unfelled μ (felled) = 48.73 μ (unfelled) = 47.97 SE = 0.22 *p* = 0.0009 ** | Felled = Unfelled μ (felled) = 46.31 μ (unfelled) = 46.70 SE = 0.32 *p* = 0.83 |
| **Nitrogen** | Felled > Unfelled μ (felled) = 0.86 μ (unfelled) = 0.74 SE = 0.03 *p* < 0.0001 ** | | Felled > Unfelled μ (felled) = 0.98 μ (unfelled) = 0.73 SE = 0.05 *p* < 0.001 ** | Felled = Unfelled μ (felled) = 0.62 μ (unfelled) = 0.56 SE = 0.032 *p* = 0.047 * |
| **Lignin** | Felled < Unfelled μ (felled) = 13.36 μ (unfelled) = 14.39 SE = 0.0.42 *p* < 0.0035 ** | | Felled < Unfelled μ (felled) = 13.50 μ (unfelled) = 15.74 SE = 0.81 *p* = 0.0037 ** | Felled > Unfelled μ (felled) = 16.46 μ (unfelled) = 16.37 SE = 0.70 *p* = 0.51 |
| **Salicortin** | Felled > Unfelled μ (felled) = 3.83 μ (unfelled) = 2.55 SE = 0.41 *p* = 0.002 ** | Felled > Unfelled μ (felled) = 1.39 μ (unfelled) = 0.93 SE = 0.15 *p* = 0.002 ** | Felled > Unfelled μ (felled) = 11.62 μ (unfelled) = 8.0 SE = 1.04 *p* = 0.0003 ** | Felled = Unfelled μ (felled) = 1.98 μ (unfelled) = 2.35 SE = 0.49 *p* = 0.06 |
| **HCH-Salicortin** | Felled > Unfelled μ (felled) = 0.55 μ (unfelled) = 0.39 SE = 0.06 *p* = 0.009 ** | Felled > Unfelled μ (felled) = 0.37 μ (unfelled) = 0.26 SE = 0.06 *p* = 0.035 * | Felled = Unfelled μ (felled) = 0.35 μ (unfelled) = 0.44 SE = 0.17 *p* = 0.29 | Felled = Unfelled μ (felled) = 0.60 μ (unfelled) = 0.20 SE = 0.15 *p* = 0.13 |
| **Condensed Tannins** | Not Tested | | Felled = Unfelled μ (felled) = 2.02 μ (unfelled) = 1.85 SE = 0.14 *p* = 0.12 | Not Tested |

For Fremont cottonwood in Arizona, our hypothesis that arthropod species richness and abundance would differ significantly between felled and unfelled trees at each site was also supported (Table 2). For the August 2009 sampling period, arthropod species richness was 56.9% greater in felled trees at Dry Beaver Creek than in unfelled trees. Arthropod richness at the Verde River showed this same pattern in August 2009, with felled trees supporting 38.3% greater arthropod richness than unfelled trees. Relativized arthropod abundance on resprout growth of felled trees was 92.3% greater at Dry Beaver Creek and 114.4% greater on the Verde River than relativized abundance on juvenile growth of unfelled trees in August 2009. Total abundance did not show significant differences. Shannon's Diversity at Dry Beaver Creek was 1282% higher on felled than on unfelled trees,

and at the Verde River, Shannon's Diversity was 31.7% higher on the resprout growth of felled trees than on paired unfelled trees. With the rigor of Bonferroni adjustments, the May 2009 sampling period only showed a significant difference with respect to Shannon's Diversity, with felled trees at the Dry Beaver Creek site supporting 52% higher species diversity than unfelled trees. For all other metrics, there was no significant difference between treatments, though 3 analyses (total abundance at the Verde River; relativized abundance and species richness at Dry Beaver Creek) were significant before Bonferroni adjustments. Importantly, there were no cases of the juvenile growth of unfelled trees supporting significantly greater richness, abundance or diversity than resprout growth on felled trees.

**Table 2.** Arthropod surveys on juvenile growth of Fremont cottonwood at both Arizona sites and on *narrowleaf cottonwood* on the San Miguel River generally showed greater relativized abundance, species richness and diversity in resprout growth of felled trees than in unfelled control trees. Eastern cottonwood on the San Miguel River showed no responses. Directionality of each difference, if any, along with the corresponding *p*-value is reported. All analyses were conducted using a non-parametric Wilcoxon-Signed Rank Test, and Bonferroni adjusted for an α of 0.008 for all sites. *p*-values with an asterisk * are significant prior to Bonferroni-adjustments; *p*-values with two asterisks ** are significant with Bonferroni-adjusted α. With Fremont cottonwood, we found significant differences in 10 out of 16 analyses without Bonferroni adjustments and 7 out of 16 analyses when using Bonferroni adjustments. In narrowleaf cottonwood, we found significant differences in 3 out of 4 analyses without Bonferroni adjustments and 1 out of 4 analyses with Bonferroni adjustments. No significant differences were found in 4 analyses with Eastern cottonwood.

| Metric | Date | Site | | Date | | |
| :---: | :---: | :---: | :---: | :---: | :---: | :---: |
| | | **Dry Beaver Creek, AZ Fremont** | **Verde River, AZ Fremont** | | **San Miguel, CO Narrowleaf** | **San Miguel, CO Eastern** |
| | | Results | Results | | Results | Results |
| **Total Abundance** | May 2009 | $N$ = 19 pairs<br>Felled = Unfelled<br>μ (felled) = 14.58<br>μ (unfelled) = 28.73<br>SE = 10.53<br>$p$ = 0.91 | $N$ = 30 pairs<br>Felled > Unfelled<br>μ (felled) = 23.9<br>μ (unfelled) = 8.93<br>SE = 11.44<br>$p$ = 0.013 * | May/June 2009 | $N$ = 25 pairs<br>Felled > Unfelled<br>μ (felled) = 23.4<br>μ (unfelled) = 9.32<br>SE = 9.84<br>$p$ = 0.004 ** | $N$ = 25 pairs<br>Felled = Unfelled<br>μ (felled) = 29.50<br>μ (unfelled) = 41.71<br>SE = 9.38<br>$p$ = 0.23 |
| **Relativized Abundances** | | $N$ = 19 pairs<br>Felled > Unfelled<br>μ (felled) = 3.25<br>μ (unfelled) = 2.40<br>SE = 0.39<br>$p$ = 0.019 * | $N$ = 30 pairs<br>Felled = Unfelled<br>μ (felled) = 2.60<br>μ (unfelled) = 2.02<br>SE = 0.43<br>$p$ = 0.11 | | $N$ = 25 pairs<br>Felled > Unfelled<br>μ (felled) = 3.62<br>μ (unfelled) = 2.66<br>SE = 0.50<br>$p$ = 0.02 * | $N$ = 25 pairs<br>Felled = Unfelled<br>μ (felled) = 1.46<br>μ (unfelled) = 1.81<br>SE = 0.33<br>$p$ = 0.13 |
| **Species Richness** | | $N$ =19 pairs<br>Felled > Unfelled<br>μ (felled) = 4.74<br>μ (unfelled) = 3.74<br>SE = 0.47<br>$p$ = 0.026 * | $N$ = 30 pairs<br>Felled = Unfelled<br>μ (felled) = 4.50<br>μ (unfelled) = 3.63<br>SE = 0.57<br>$p$ = 0.08 | | $N$ = 25 pairs<br>Felled > Unfelled<br>μ (felled) = 5.8<br>μ (unfelled) = 4.6<br>SE = 0.59<br>$p$ = 0.038 * | $N$ = 22 pairs<br>Felled = Unfelled<br>μ (felled) = 2.95<br>μ (unfelled) = 3.27<br>SE = 0.48<br>$p$ = 0.69 |
| **Shannon's Diversity** | | $N$ = 19 pairs<br>Felled > Unfelled<br>μ (felled) = 1.14<br>μ (unfelled) = 0.75<br>SE = 0.13<br>$p$ = 0.0039 ** | $N$ = 30 pairs<br>Felled = Unfelled<br>μ (felled) = 1.09<br>μ (unfelled) = 0.97<br>SE = 0.16<br>$p$ = 0.08 | | $N$ = 25 pairs<br>Felled = Unfelled<br>μ (felled) = 1.44<br>μ (unfelled) = 1.31<br>SE = 0.13<br>$p$ = 0.20 | $N$ = 23 pairs<br>Felled = Unfelled<br>μ (felled) = 0.63<br>μ (unfelled) = 0.72<br>SE = 0.11<br>$p$ = 0.34 |
| **Total Abundance** | August 2009 | $N$ = 18 pairs<br>Felled = Unfelled<br>μ (felled) = 23.94<br>μ (unfelled) = 15.94<br>SE = 5.30<br>$p$ = 0.08 | $N$ = 28 pairs<br>Felled = Unfelled<br>μ (felled) = 53.57<br>μ (unfelled) = 37.79<br>SE = 14.82<br>$p$ = 0.13 | | N/A | N/A |

**Table 2.** *Cont.*

| | | Site | | | | |
| Metric | Date | Dry Beaver Creek, AZ Fremont | Verde River, AZ Fremont | Date | San Miguel, CO Narrowleaf | San Miguel, CO Eastern |
| | | Results | Results | | Results | Results |
| **Relativized Abundance** | | *N* = 18 pairs<br>Felled > Unfelled<br>μ (felled) = 2.75<br>μ (unfelled) = 1.43<br>SE = 0.38<br>*p* = 0.0015 ** | *N* = 28 pairs<br>Felled > Unfelled<br>μ (felled) = 3.43<br>μ (unfelled) = 1.60<br>SE = 0.37<br>*p* < 0.0001 ** | | N/A | N/A |
| **Species Richness** | | *N* = 18 pairs<br>Felled > Unfelled<br>μ (felled) = 7.06<br>μ (unfelled) = 4.50<br>SE = 0.56<br>*p* = 0.0003 ** | *N* = 28 pairs<br>Felled > Unfelled<br>μ (felled) = 7.08<br>μ (unfelled) = 5.12<br>SE = 0.61<br>*p* = 0.0018 ** | | N/A | N/A |
| **Shannon's Diversity** | | *N* = 18 pairs<br>Felled > Unfelled<br>μ (felled) = 1.52<br>μ (unfelled) = 0.11<br>SE = 0.56<br>*p* = 0.0007 ** | *N* = 27 pairs<br>Felled > Unfelled<br>μ (felled) = 1.58<br>μ (unfelled) = 1.20<br>SE = 0.12<br>*p* = 0.0029 ** | | N/A | N/A |

*Indicator species*—We found three arthropod species served as indicator species of beaver-felled trees in Arizona Fremont cottonwoods, and one species that was an indicator of unfelled trees in Colorado narrowleaf cottonwoods. At the Verde River in August 2009, two leaf modifiers were indicative of resprout growth on felled trees: *Anacampsis niveopulvella* Chambers, 1875 (order Lepidoptera, family Gelichiidae, Figure 1B) had an indicator value of 33.2% (*p* = 0.0286), and a species of clearwing moth in the order Lepidoptera and family Sesiidae had an indicator value of 41.6% (*p* = 0.0412). At Dry Beaver Creek, a leafroller moth in the order Lepidoptera and family Tortricidae (Figure 1C) had an indicator value of 27.8% (*p* = 0.0488) on beaver-felled trees.

Colorado Sites: Our MRPP/NMDS analyses of narrowleaf and Eastern cottonwood trees on the San Miguel River supported the hypothesis that arthropod communities would differ between study locations as well as between the juvenile resprout growth of felled and unfelled trees for narrowleaf cottonwoods (A = 0.015, *p* = 0.0002), but not for Eastern cottonwoods (A = −0.00099, *p* = 0.52).

For narrowleaf cottonwood on the San Miguel River, surveys again showed that beaver herbivory generally had a positive impact on arthropod abundance (total abundance, *p* = 0.004), but no significant difference in species richness (*p* = 0.038, significant prior to Bonferroni adjustment) or Shannon's Diversity (*p* = 0.20) (Table 2). Resprout growth on felled narrowleaf trees supported 115.1% higher total abundance than juvenile growth on unfelled trees. We found no evidence in narrowleaf of unfelled trees supporting significantly greater arthropod richness or abundance than resprout growth on felled trees.

*Indicator species*—In narrowleaf cottonwood on the San Miguel River, one species of ant (*Formica propinqua* Creighton, 1940, order Hymenoptera, family Formicidae) was indicative of unfelled trees, with an indicator value of 58.3% (*p* = 0.0122).

## 4. Discussion

### 4.1. Feeding Preferences for Cottonwoods over Other Native and Exotic Species

Our hypothesis that beavers would exhibit feeding preferences for different woody vegetation was strongly supported. When given a choice in cafeteria experiments, beavers chose Fremont cottonwood six times more often than any other locally abundant species including native Arizona sycamore, velvet ash, and coyote willow and highly invasive exotic tamarisk species. These findings are important because they show that cottonwoods

and beavers can be very strong interactors such that beaver preferences for this tree have the potential to influence the evolution of tree phytochemical defenses, alter competitive interactions with less preferred species, and affect the associated arthropod community.

Although we did not perform the same experiment with narrowleaf and Eastern cottonwood, these findings are in agreement with other studies, showing that the selective foraging of beavers can change riparian forest composition. Lesica and Miles [48] found that beaver chose Eastern cottonwood over exotic tree species in Eastern Montana. In an observational study that monitored beaver felling of trees along the Weber River in northern Utah, Bailey et al. [49] found that in just two years, cottonwoods declined by 17–21%, native willow increased 17%, and cumulatively exotic Russian olive (*Elaeagnus angustifolia* L.) and tamarisk increased by 4–17%, depending on the location assessed (i.e., stream margin or inner gallery forest). At an even finer scale, cafeteria experiments have shown that beavers prefer individual narrowleaf cottonwood genotypes low in condensed tannins and avoid those high in condensed tannins [15].

### 4.2. Phytochemical Responses That May Deter Future Beaver Herbivory

In support of our hypothesis that beavers would affect the phytochemistry of resprout growth relative to control juvenile growth of unfelled trees, we found that twig tissue of resprout growth of Fremont and narrowleaf cottonwood was generally significantly higher in nitrogen, carbon, salicortin, and HCH-salicortin than that of control trees, which could affect the suitability of resprout growth for subsequent beaver foraging. Additionally, twigs from resprout growth of felled Fremont and narrowleaf cottonwood trees generally had significantly lower concentrations of lignin than those of unfelled trees. In Fremont cottonwood, we found significant differences between the resprout growth of felled trees and juvenile growth of unfelled trees in five out of seven analyses when using Bonferroni adjustments and in six out of seven analyses without Bonferroni adjustments. In narrowleaf cottonwood, we found significant differences in four out of six analyses, all of which remained significant with Bonferroni adjustments. In combination, these results may have important implications for beavers' selection of trees with certain chemical profiles [50]. As nitrogen is an essential dietary nutrient, beavers could prefer trees with elevated concentrations of this element. In contrast, as lignin is an indigestible structural compound in woody plants, beavers might avoid trees with high lignin concentration. Similarly, phenolic glycosides such as salicortin are thought to deter mammalian herbivory as they do in elk [51]. Aspen (*P. tremuloides*) saplings with a 2% concentration of the phenolic glycoside tremulacin suffered 80% mortality by elk, whereas trees with an 8% concentration suffered only 5% mortality. Increasing tremulacin concentrations in browsed but surviving saplings suggested that elk herbivory induces chemical resistance in these trees.

Cafeteria experiments performed by Basey et al. [52,53] showed that aspen resprout growth following beaver felling of trees was highly avoided by beavers relative to controls. They also found that an unidentified phenolic compound was about 18 times greater in the avoided resprout growth, which strongly suggests that the induction of phytochemical defenses deterred subsequent beaver herbivory. Thus, the results we observed of elevated production of defensive chemicals in resprout growth were also likely the results of induction and may be adaptive in deterring future beaver herbivory, as seen in studies that found a relationship of reduced feeding by porcupine (*Erithizon dorsatus* Linnaeus, 1758) and elk (*Cervus elaphus* Lannaeus, 1758) when aspen have elevated levels of phenolic glycosides such as salicortin and HCH-salicortin [54,55]. However, in order to reap the benefits of elevated nitrogen concentration in highly defended resprout, it has been suggested that beavers may utilize a behavioral strategy of leaching freshly cut twigs in food caches near dams and lodges to remove undesirable phenolics [56].

In contrast to the above significant findings with two species of *Populus*, Eastern cottonwood showed a significant difference between felled and unfelled trees only with respect to nitrogen (one of five analyses), but this significance did not hold up under Bonferroni adjustments. Since the sample sizes and methods were the same as our other

studies, it is unclear why this cottonwood responded differently. Future studies could explore seasonal patterns and potentially utilize these differences to better understand the causal mechanisms of induction.

### 4.3. Phytochemical Responses to Herbivory That Affect Arthropods

Although our studies of twig chemistry do not allow us to evaluate induced responses on foliar feeding insects, other studies have examined the effect of beaver on a specialized beetle. In a common garden with clonal replicates of the same tree genotypes, Martinsen et al. [12] found that resprout foliage from the felled clones was significantly higher in salicortin (100% greater), and total nitrogen (20% greater) than paired unfelled control clones. These differences had a major positive impact on the distribution of the beetle *Chrysomela confluens* Rogers, 1856, which sequesters the plant's defenses for its own. Beetles were 15 times more abundant on resprout than control growth where their own defenses were elevated, and experiments showed that beetle larvae raised on resprout growth were more successful in deterring predaceous ants than those raised on control growth.

Previous research has shown that herbivory induces defensive (i.e., tannins and phenolic glycosides) and nutritional (i.e., carbon and nitrogen) chemistry in woody plants [12,52,53,57,58]. With narrowleaf cottonwood, Body et al. [59] found that the presence of the gall-forming aphid *Pemphigus betae* Doane, 1900, triggered the induction of 15 different phytohormones belonging to 5 different classes. The concentrations of these phytohormones in both constitutive and induced responses in galled leaves were tree genotype-specific and exhibited high broad-sense heritability that ranged from 0.39 to 0.93 and from 0.28 to 0.66, respectively. With such changes in phytochemistry in the presence of gall aphids, Keith et al. [6] found that when these aphids were experimentally removed, the composition and network structure of an arthropod community of 139 species were greatly affected. Thus, changes in leaf phytochemistry with herbivory may be one mechanism driving differences in arthropod abundance, richness, and community structure [60]. If some trees are better defended than others, generalist herbivores should be repelled, but specialists that use these chemicals for their own defenses or can otherwise tolerate high levels of phenolic glycosides should be abundant on highly defended resprout growth.

### 4.4. Interacting Foundation Species (Beavers and Cottonwoods) Redefine Communities

Our hypothesis that beaver felling of trees causes resprout growth of felled trees to support a different community of arthropods was generally confirmed for two of the three cottonwoods studied. In all four analyses, Eastern cottonwood showed no significant differences between felled and unfelled trees. Fremont supported greater arthropod abundance, species richness and diversity in 10 out of 16 analyses, 7 of which retained significance with Bonferroni adjustments; narrowleaf exhibited the same pattern in 3 out of 4 analyses, 1 of which was significant with Bonferroni adjustments. In no analysis did we find significant differences in the opposite direction, in which resprout growth supported reduced metrics of community structure.

We propose that these differences should be expected when two or more interacting foundation species strongly interact. Studies examining interactions between foundation species have shown that one or two influential species can affect ecological communities and biodiversity at a large scale [5,9,11,14,42,61]. For example, multiple studies have shown the dramatic effect sea otters (*Enhydra lutris* Linnaeus, 1758) have on populations of sea urchins *Strongylocentrotus* spp. and how this cascades to affect the abundance and diversity of kelp forests [62,63]. Cottonwood–beaver interactions reveal another interacting foundation species, the leafroller moth *Anacampsis niveopulvella*, which is an indicator species of resprout foliage (Figure 1C). Studies by Martinsen et al. [64] found that the leaf roll produced by this moth created a shelter that attracted many other arthropods, in which the experimental removal of leaf rolls caused a five-fold decline in species richness and a seven-fold decline in abundance relative to controls with leaf rolls. In an opposite example, Busby et al. [65] found that with the addition of the leaf pathogen *Drepanopeziza populi*

(Lib.) Rossman & W.C.Allen to Fremont and narrowleaf cottonwood and their naturally occurring hybrids, arthropod species composition, richness and abundance were negatively affected both within and among plant species. In the above examples, the interaction of two foundation species affected a much larger arthropod community but the signs of the interactions were different. In the moth–tree interaction, the effect was positive, whereas with the pathogen–tree interaction, the effect was negative at the individual tree level. The change in the sign of the interaction is likely due to the positive effects of leaf rolling moths by creating shelters for other arthropods. In contrast, the leaf pathogen destroys leaf tissue eliminating food resources for other species resulting in a negative effect on most arthropods. However, in both cases, at the stand level, the mix of trees with and without interactions enhanced overall arthropod diversity because the presence and absence of interactions between foundation species supported different communities.

*4.5. Conservation and Management*

Beavers were historically present in most North American streams but have been reintroduced and are naturally colonizing many streams after almost complete extirpation in the late 1800s [27]. Because of the important role they play in structuring riparian environments and regulating ecosystem processes [11,66], beavers are integral to healthy riparian systems, and it is important to understand and utilize their diverse effects in land management [67,68]. In addition to their positive effects on biodiversity, they have been credited with preserving critical habitats essential to threatened species. Bartel et al. [69] found that beavers indirectly maintain populations of rare butterfly *Neonympha mitchellii ssp. francisci* Parshall & Kral, 1989, by modifying the composition and diversity of plant communities within wetlands. Similarly, Bouwes et al. [70] found that natural and simulated beaver dams increased the quantity and complexity of habitats of a threatened population of steelhead *Oncorhynchus mykiss* Walbaum, 1792.

On the negative side, the selective foraging of beavers on cottonwoods can promote exotic and invasive species. Our cafeteria experiment (Figure 2) showed that beavers are strong interactors with Fremont cottonwood and avoided exotic tamarisk, which is especially invasive in the American West. In an observational study that monitored beaver felling of trees along the Weber River in northern Utah, Bailey et al. [49] found that in just two years, cottonwoods declined by 17–21%, native willow increased by 17%, and cumulatively exotic Russian olive and tamarisk increased by 4–17%, depending on the location assessed (i.e., stream margin or inner gallery forest). Similarly, in Montana, Lesica and Miles [48] found that in stands where beaver had been present, an average of 80% of cottonwood trees had been felled, while only rarely did they fell Russian olive or tamarisk. Thus, these three studies conducted over nearly 750 km or 7° of latitude agree that selective foraging of beavers promote highly invasive species. This switch from native to exotic species is also associated with changes in the arthropod and avian communities [71–73] and beaver preference for native cottonwoods (Figure 2) facilitates the invasion of exotic species such as tamarisk and Russian olive (see also [48,49]). To avoid further conversion to exotic species, managers should remove exotics and protect native cottonwoods from beavers through fencing or other non-destructive means. One reviewer points out that this is not a beaver problem but, rather, a human problem as humans have introduced these exotic species. While this is correct, it emphasizes that human introduction of exotic species can easily be compounded by native herbivores through their selective avoidance of exotic species that are difficult and costly to remove. We emphasize that in systems lacking these invasive plants, beavers have been shown to positively influence biodiversity [7,25] and habitat heterogeneity [74]. Therefore, when coupled with the removal of exotics, preserving riparian areas with resident beaver populations should facilitate an overall preservation of complex interactions and diverse communities.

Not only is it important to understand how these critical mammalian herbivores interact with other riparian community members to enhance biodiversity, but perceptions by landowners will also greatly impact management decisions. Charnley et al. [68] reported

that in the United States, human perceptions of beavers are highly variable. Where ranching is not dominant, beavers are often viewed as more of a nuisance species than as an asset, whereas in the northwest states, the majority of ranchers viewed the benefits of beavers and their dams as being greater than their drawbacks. One Idaho rancher stated "It worked well for everything because one, it provided water, year-round water all the time, which is a godsend for wildlife, for my cattle, everything. Two, it enhanced the wet meadows that were there, so you had better forage production for cattle, wildlife, everything else" [68]. Beavers and their dam building behavior may also play an important role in climate change mitigation. Since the American Southwest is currently in an ongoing 19-year megadrought that is considered the 2nd worst in 1200 years [75], the retention of water by beaver dams may be an important climate change adaptation strategy for ranchers [68]. Fairfax and Whittle [76] found that the higher water table and wetting resulting from beaver-dammed riparian corridors were relatively unaffected by wildfires compared to riparian corridors without beaver damming. Thus, it appears that land managers can use beavers to mitigate climate change impacts on riparian ecosystems [77].

## 5. Conclusions

Our study found patterns largely in agreement with our hypotheses that beavers preferentially choose cottonwoods over other tree species, impacting cottonwood twig chemistry and influencing arboreal arthropod communities: (1) When offered a variety of locally occurring woody plant species, beavers preferentially selected Fremont cottonwood six times more often than other available food samples; (2) Twig tissue from resprout growth of felled trees showed different defensive, structural and nutritional phytochemistry concentrations than twig tissue from paired unfelled trees; and (3) Juvenile resprout growth on beaver-felled trees supported significantly different arthropod communities, often with higher species richness, diversity and abundance than the juvenile growth of unfelled cottonwoods. This trio of patterns was repeated in both Fremont and narrowleaf cottonwood stands, arguing that cottonwood stands with heterogeneity resulting from beaver felling support higher arthropod diversity than cottonwood stands without beaver influence. These alterations to the phytochemistry of juvenile resprout growth following beaver felling may impact future herbivory. Strikingly, Eastern cottonwood did not follow these patterns, and we are unaware of studies that have investigated reasons for this different response. Our findings indicate that as interacting foundation species, beavers and cottonwoods can impact the phytochemistry of riparian tree stands as well as the community structure of arthropods on these trees. As riparian habitat becomes ever more imperiled due to climate change, invasive species, and development, our understanding of these interactions and their community-wide impacts are increasingly critical for conservation and management. As we further our understanding of these complex and fragile ecosystems, we can be better prepared to prioritize research and responses that protect and restore functionality for biotic communities and ecosystem processes.

**Author Contributions:** Conceptualization, T.G.W., R.M.D., A.R.K., S.R.H., Z.K. and S.M.S.; data curation, R.M.D.; formal analysis, R.M.D. and S.M.S.; funding acquisition, T.G.W. and S.M.S.; investigation, R.M.D., A.R.K. and S.R.H.; methodology, R.M.D., L.H. and R.L.L.; project administration, R.M.D.; supervision, T.G.W. and S.M.S.; visualization, S.M.S.; writing—original draft, R.M.D.; writing—review and editing, T.G.W., R.M.D., F.M.W., S.R.H., Z.K., R.L.L. and S.M.S. All authors have read and agreed to the published version of the manuscript.

**Funding:** This research was funded, in part, by NSF FIBR grant DEB-0425908, Macrosystems grants DEB-1340852 and DEB-2017877, and BEE grant DEB-1914433.

**Institutional Review Board Statement:** Not applicable.

**Informed Consent Statement:** Not applicable.

**Data Availability Statement:** The data presented in this study are openly available in FigShare at https://doi.org/10.6084/m9.figshare.14692761, (accessed on 27 May 2021).

**Acknowledgments:** We thank Tad Theimer, the Shuster and Whitham lab groups for providing valuable feedback and discussion. Many thanks to Ryan Paulk, Whitney White, Helen Bothwell, and Chase Ridenour for assistance in the field. Adrian Stone and Matt Zinkgraf provided assistance with statistical analysis.

**Conflicts of Interest:** The authors declare no conflict of interest. The funders had no role in the design of the study; in the collection, analyses, or interpretation of data; in the writing of the manuscript, or in the decision to publish the results.

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
