# Peer review of "Beavers, Bugs and Chemistry: A Mammalian Herbivore Changes Chemistry Composition and Arthropod Communities in Foundation Tree Species"

_forests, doi:10.3390/f12070877_

Round 1
Reviewer 1 Report
There has been an improvement on the work done. However, I had suggested introducing a list of the arthropods used in calculations presented in Table 2, and that did not happen, perhaps because the author did not agree with my opinion. I introducing this information again.
Author Response
In response to the reviewer's comment:
"There has been an improvement on the work done. However, I had suggested introducing a list of the arthropods used in calculations presented in Table 2, and that did not happen, perhaps because the author did not agree with my opinion. I introducing this information again."
We feel the list of arthropods is too long for inclusion in the table text but have included spreadsheets in FigShare, which include all arthropods used in calculations.
Reviewer 2 Report
I like the concept of foundation species interactions and ecosystem function – great model to test. General comments center around the ms. needing just a bit more care in editing (for examples, see specific comments 1, 2, 6, 11, 17, 18, and others), the way the hypotheses are written (see comment 10), inconsistencies in methods regarding paired trees used for chemistry and arthropod surveys (comment 15), results presentation (see comments 17, 18, 19), and how the arthropod survey data are discussed. I’m not convinced that the cafeteria study with only one species of cottonwood supports the main premise of the ms. If you had experiments for all cottonwood species, then it would make more sense. The fact that you do not present any foraging data at the Colorado sites you are asking the reader to assume that beavers take all cottonwood species based on their selection of one species. Beavers are choosy generalist and use many species of woody vegetation while showing preference for a few. However, they do show preferences for specific Betula species and Acer species (and Salix species) and not others. Without data on preferences for eastern and narrowleaf cottonwoods the case is weaker. To me the cafeteria experiment, while interesting, does not add much to your ms.
The differences between the three cottonwood species and their responses to beaver herbivory are very interesting. A nice study would be to compare pure stands of eastern and narrowleaf with hybrid stands and look at intensity of beaver foraging in each stand and the chemical and arthropod impacts. The hypothesis would be that beavers influence narrowleaf chemistry and arthropod diversity and abundance but not eastern and that there should be intermediate responses in the hybrids. It would be important to have a good handle on availability at each site and intensity of foraging by beavers at each site. Just a thought for future work.
Specific comments:
- Abstract: Line 14 - Please use the common name “North American beaver” not just American beaver. While common names are always problematic and C. canadensis has been called both the Canadian and American beaver the most appropriate common name is “North
American beaver.”
- Line 18 – There is a grammatical problem here; you write, “…we studied preferred food resources of the beaver on P. fremontii…” This does not make sense. I think you mean something like this:
“We experimentally examined the impact of beaver foraging on riparian communities by first investigating beaver food preferences for one cottonwood species, P. fremontii, compared to other locally available woody species. We next examined the impact of beaver foraging on twig chemistry and arthropod communities in paired samples of felled and unfelled cottonwood species in northern Arizona (P. fremontii) and southwestern Colorado (P. angustifolia, P. deltoides).
- Lines 27 – Would you consider “The above findings “suggest” instead of argueand maybe rethink what this sentence really says. You only found chemical and arthropod differences in two of the three cottonwood species so maybe what you write is a bit too inclusive?
- Line 30 – Would you consider something like this, “Our results indicate that beaver herbivory changes the chemical composition in two of the three cottonwood species (P. Fremontii and P. angustifolia) potentially allowing better defense against future beaver herbivory.”
I left out the ontogeny of chemistry and the part about being similar to juvenile growth. Why? If I read your paper correctly it is only juvenile growth from beaver cut trees that show chemical and arthropod differences and what you compare these to is unfelled juvenile growth. Thus, your results - if you found differences as you report, your results can’t be similar to juvenile growth of unfelled trees, which is what you write here. Your results illustrate differences in some chemistry between felled and unfelled juvenile growth.
- Introduction: Page 2: Line 50 – I think you want to write species not “organisms” here since you don’t mean that there is individual organism variation that makes one individual more of a fundamental player than another within the same species (although that is a very interesting question) but it is the species that plays a fundamental role in the ecosystem?
- Line 61 – “Global change”? Do you mean anthropocentric habitat and climate change? Global change seems a bit vague of a term here.
- Line 63 - Would you consider, “…foundation species hypothesis, “and found that” the interactions…”
- Line 68 – What you say is fine, but just reads a bit awkward. Maybe, “By felling trees that are used for food and construction activities (building dams that slow water flow and create ponds, and lodges), beavers …”
- Page 3, line 93-102. This paragraph reads like methods more than introduction. Do you really need this paragraph? I’d suggest omitting it and just get on to your hypotheses. Why not cut it out and see how it reads? Also, Figure 1B is of juvenile growth on a felled tree and in lines 93 – 94 it reads like 1B shows both felled regrowth and unfelled regrowth. It may be helpful to add a picture of what juvenile growth on an unfelled cottonwood looks like. It is also interesting that you show narrowleaf regrowth when this is the species you examined that did not support your hypotheses.
- Page 3: Lines 106-112 – I like that you have hypotheses here, but think they can be written in a bit better form. For example, hypothesis 1. Is your hypothesis really that beavers exhibit feeding preferences. If so you really only a found significant preference for P. fremontiii and no other species so do you accept or reject your hypothesis? Also, given the vast literature on beavers being “choosy generalist herbivores” this hypothesis really does not say much. I think your real hypothesis was that, “Beavers will show a preference for P. fremontii over thee other native and one invasive woody species common to the area. This seems appropriate since beavers are known to favor aspen (another species of Populus) when available in their habitat.” I’m not familiar with the willow species but was surprised they did not also take this. I wonder if the natural availability of the species you used in your study had anything to do with the selection? Also, Cottonwood grows very much like a large tree=, while some of the other species you used grow more shrub like (I assume the willow species is not a tree). Could this have influenced your study?
My major point for all your hypotheses is that they could be a bit more developed. Especially since your results only partially allow acceptance of the hypotheses.
- Line 113 – Hypotheses are not questions but statements. Thus, they do not provide “answers” Why not write, “Testing these hypotheses increases our knowledge…”
- Methods: Pages 3-4: up to line 144. I’m not convinced that you need the introductory sentences to the study sites (about what your data collection). As far as the Arizona sites you mention that Dry Beaver Creek experiences beaver herbivory during annual spring flooding (lines 130-131) but your cafeteria study was during October and November? There must have been beavers there or they would not have eaten the branches you left out. Maybe you need to explain that DBC is an intermittent tributary and beavers create ponds that persist during the dry(er) months so the reader knows that beavers live there permanently. Just a bit more here would help. Additionally, page 4, line 132, says no narrowleaf cottonwood occurs in AZ but does not say anything about eastern cottonwood? Would it be better to write that since only P. fremontii grows along the San Miguel River and since you wanted to test the impact of beaver herbivory on other cottonwood species you chose the sites in Colorado. As an aside, having a hybrid zone seems like a great opportunity (maybe missed?) to test the two species and how the hybrids react to beavers??
Line 134 – just be careful here since you only tested the chemistry and arthropod hypotheses in CO – no foraging study. So you only tested hypotheses 2 and 3 at your Colorado sites. As you write it you tested all of your hypotheses here.
- Cafeteria study: Maybe to help out readers not familiar with native and exotic species in AZ you could just include which are native and which are not (you do have this somewhere in the paper but I’d put it here. You really only have 1 non-native species so buy writing that you tested native and exotic species you are technically correct but maybe overstating the “exotic species” just a bit. Why not be up front and say you used readily available (or the five most abundant??) woody species, 4 native and 1 non-native). I assume that you removed all uneaten branches after trial 1 and replace them with new branches for trial 2? This could be clearer.
- Chemical analyses: Line 162 – You write “these trees”? I think it is better to say that, “To analyze the chemical profiles of the three cottonwood species we collected 45 cm of the terminal branch growth from the juvenile zone of a sample of trees at each site.”
- Line 170-171 – Note that you indicate that you used the same trees for chemical studies and Arthropod surveys, which is great. However, if I read this correctly you collected branches from 24 pairs of trees at each AZ site for chemistry. For Arthropod surveys you indicate you surveyed 30 pairs at the VR site. This does not add up since you only had 24 pairs for chemistry. If you surveyed additional pairs you need to add that in (and that the assumptions is that the chemistry in those 6 other pairs would be similar). Also note that you used 24 pairs at DBC for chemistry but only 19 for Arthropod surveys. Also note that the number used for chemistry and arthropod surveys do not add up in CO either. This needs to be clear. Your data is stronger and more convincing if you surveyed the same trees for arthropods as you examined for chemistry. If this is what you did you can actually compare individual tree chemistry with arthropod abundance and not just rely on averages.
- Line 197 – I think it is more correct to say that beaver herbivory may alter tree chemistry, which in turn may influence arthropod abundance. What you write is probably okay, but I really think (and you discuss this later) that it is the change in resprout chemistry that supports the different arthropod community. Just a thought.
- Results: Figure 3. Why do you mention felled and unfelled trees at the AZ sites but do not show these in 3A? Shouldn’t there be felled and unfelled at each site (DBC and VR) in the figure? Also, The way 3A is presented the sites do not appear significantly different? You need to add felled and unfelled to 3A or change the caption to include why it is not presented.
- Table 1. Why are C, N, and lignin not separated by site. If there is some reason you combined the sites for these analyses you need to write this in the methods. Note: you have the arrow reversed for nitrogen in AZ sites (felled is 0.74 unfelled is 0.86 so felled<unfelled. Your table indicates that nitrogen is lower in P. fremontii in felled juvenile growth. I don’t see the 16.2% higher nitrogen in felled trees here. Based on Table 1 I’d say that unfelled juvenile growth is 14% higher in nitrogen than felled juvenile growth. Can you clear this up?
- General comment about results. You really need to make sure that a reader has no questions about the data you present. If you refer to a figure as showing felled and unfelled results it show clearly show it. If nitrogen content for P. fremontii is higher in felled regrowth then your table should show it. Why data for the two AZ sites appears to be lumped for N, C, and lignin needs to be explained. Otherwise it is difficult for the reader to see clearly the significance of your work in understanding ecosystems and foundation species interactions.
- Discussion: Page 12, line 420-421 – Table 1 indicates that nitrogen is higher in unfelled trees in AZ.
- line 479-480 – I think you need to say that beaver herbivory influences regrowth chemistry, which then influences the arthropod community. Also note that here you indicate that in “only” 8/24 sampling periods (across all sites?) – 33% - the arthropod community was greater. This means that 66% of the time it was not different on felled or unfelled trees. Somehow I missed this in the results.
- Line 486 – You need to add “felled trees” in here. 8/24 or 13/24. Either way I’m not sure these are significant results to support your hypothesis. It must be how you include this idea of “sampling period” here, which seems to contradict what table 2 reports (data from paired trees). Can you clarify this?
- Lines 539-558: Conservation and management. There is nothing incorrect with what you write but I wonder if you could make a stronger statement about beavers and invasive species? It is not the evolved foraging behavior of beavers with their native species that encourages the advancement of invasives. It is the impact of humans on the landscape and their introduction of these invasive tree species that is the problem. Beaver management is really human management. The overall impact of beavers on western riparian systems is much more positive than the “encouragement” of invasive tree species. Humans created the problem and managing beavers to solve the problem is a backwards way to approach it. I’d just like to see a stronger statement about this in this section.
Author Response
Please see the attachment
This manuscript is a resubmission of an earlier submission. The following is a list of the peer review reports and author responses from that submission.
Round 1
Reviewer 1 Report
I found this manuscript to be very interesting and add to our understanding of the interactions between an herbivore species and multiple prey (tree) species. The roles of both beavers and cottonwoods as foundation species in riparian ecosystems are significant and how they interact important. Generally the manuscript was well written, if somewhat haphazardly edited. That is easy to correct. Let me offer some broad and specific comments to consider.
- Abstract: While the abstract reads well it seems to broadly generalize the findings and at times what is written is a bit at odds with the results. For example, in outlining your questions you begin on line 19 with the foraging experiments and write "cottonwoods". Actually you only conducted the cafeteria experiments with one species not all three species. Line 21 your results do not show differences in all samples or for all species. Line 27 you actually acknowledge that species of cottonwood matters. Line 30-32 where you discuss chemistry changes you mention "developmental trajectory." But I find this confusing since you measure chemistry in juvenile shoots of both felled and unfelled trees? If you measured the chemistry in juvenile shoots it does not indicate developmental trajectory but just what is present. To show trajectory wouldn't you need to compare juvenile shoots to older growth? Line 34-35 you mention completely unfelled stands but I did not see where you collected data in stands that were not used by beavers. My reading of the paper is that you chose sprouts from cut trees and uncut trees in the same stands? The point here is that the abstract should summarize what the paper will detail. I found that the abstract raised questions that were not consistently answered in the manuscript. You have good data so why not just summarize what you found?
- Introduction: Line 56 - you use the term BIOME here but only use ecosystem in the rest of the ms. I think biome is a pretty broad organizational unit. Your study looks at changes in very small parts of specific riparian systems. Line 58 - would you consider "human influenced change"? Line 63-64 here you mention communities and ecosystems. Fig.1, I like the picture (A) but do not see how B,C,D relate and you make no mention of this in the introduction. It is not until much later that you refer to Fig. 1, B (and never do for C or D). I'd move these pictures to the appropriate part of the ms. Line 87-96, reads a bit more like part of the methods section than introduction. Line 100 - You outline your four questions but for the rest of the manuscript refer to them as hypotheses. Questions are general and hypotheses are specific. Why not write them as hypotheses you are testing? Line 105 - maybe use felling not browsing. Browsing has a specific meaning in herbivory and beavers felling trees is a bit more than this.
- Methods: Line 120-125 - The description of the Arizona sites needs a bit more since you write that Dry Beaver Creek experiences beaver activity during spring, yet you will do your cafeteria experiments here in the autumn? Could you include altitudes for your sites here and maybe even a map would help the reader. You also write that there is no narrow leaf cottonwood in AZ but say nothing about eastern cottonwood? I assume that it is not in AZ either. This could use more clarity. Line 139-153: Cafeteria experiments - I'd suggest more details here (if you keep this - more on that later) since it is not clear if you ran trial 1 for 3 days, than ran trail 2 for the next 10 days or if they were run at different times (weeks apart). This makes a difference since the study was done in the autumn and this is when beavers construct their winter food caches (if they do this at your sites). Selection may change depending on the precise time. Why not include when (September, October, November and even the weeks) the trials were run? Also you use the word "sample" in multiple ways. What I believe you did was place five branches (and you are not specific of where these came from (non-beaver trees or beaver trees/old or juvenile growth) of each species around the ponds (25 branches at each pond during each trial). I think this could be much clearer. Line 155 - In describing the chemistry study you write that you collected samples from 24 pairs at the AZ sites but do not say how many from each site (12 and 12?). Also you do not describe how you assigned tree pairs until the next section. Did you use the same pairs of trees for chemistry and arthropod analysis? I hope so, and if you did why not describe the pairing of trees for both. You do go into detail on this in discussing arthropod abundance but I was left wondering how you decided on tree pairs. Are the three cotton wood species clonal like aspen? If so were the samples from the sam e clones or different clones? Nothing you can do about it now but how could you reassure a reader that there are no yearly differences in chemistry or arthropod abundance since you collected the samples from AZ and CO a year apart? Also, since you attempt to keep biomass similar between pairs (and trees) why not let the reader know what biomass you measured. I generally think your methods need clarification in many places.
- Results: Big question - do you really need the cafeteria data in this detail in this paper? It is not mentioned in your title and you only conducted it at one AZ site on one cottonwood species. I'm pretty sure that there is a wealth of literature that says beavers prefer cottonwood over the other species (I was a bit surprised by the low use of the one willow species though). Plus, if you compare what you write on line 265 and how you have described the methods, and then compare it to Figure 2 there is some inconsistency. For example, you offered 20 branches of each species (5 species or 100 total branches) and write that 10% of ash and 20% of willow were eaten (this is 2 ash branches and 4 willow branches). In Fig, 2 caption you write that 3/77 (4%) of other branches were taken. Should this be 6/80 = 7.5%? It is not a major deal but makes a reader question the study. To me, you do not need to even present the cafeteria study - it reads like an add on. You could get around this by indicating that at each site you chose there was clear evidence (since you do not present any foraging data or tree availability data) that cottonwoods were taken (and maybe use your pair data to estimate use?). If it was my manuscript I'd stick to the big point - beavers have preferred trees (note, that with no cafeteria experiments in CO you do not know what beaver selection there is) and their herbivory influences the chemistry of sprouts in felled trees and the resulting arthropod community.
Chemistry: line 277 - I'm not at all clear why you include the acceptance or rejection of your hypothesis here in the results section. Shouldn't this just be what you found and leave the acceptance or rejection for the discussion? Here I'll concentrate on the figure and table. I was confused as to why you write that there are site differences in chemistry in AZ but do not represent this in Fig. 3? You show three graphs and seem to lump the DBC and VR Arizona sites together with no explanation as to why. I would like to see the site differences. In the table 1 caption you write that chemistry composition was significantly different between felled and unfelled trees, which according to some of the data in the table is a bit of an over-statement. For carbon in AZ there is no significant difference. You should also say why you lumped Carbon, Nitrogen and lignin for the AZ sites but separated out the salicortin. Please review the table and make sure it is accurate - for nitrogen in AZ unfelled trees is greater than felled, not what you represent. I know you write that there were no significant differences for eastern cottonwoods but think you should let the reader see these data as well in the table.
Line 322 - Arthropods - Again I'm not sure why you write about hypothesis acceptance here rather than in the discussion. Table 2 caption - I'd say that the first sentence here is inaccurate as well. Look at what you report for May 2009 at DBC (for some reason you put unfelled first here) but unfelled has a higher abundance than felled. I'd also check the stats on this since that sure looks like it could be significant to me. This is curious as it reverses by August at this site. I wonder if this is a seasonal effect at this site in arthropod abundance as much as the influence of beaver herbivory?
I'd consider a new sub-heading for the indicator species section after table 2. and any reason why you have specific species for the leaf modifiers but not for the ant?
5. Discussion: Why not order the discussion in order of your hypotheses? Cafeteria (if you keep this), chemistry changes, arthropod changes. The you can discuss the foundation species interactions and the conservation and management aspects of your work. Your work raises a big question by showing increases (at least seasonally) in carbon and nitrogen in felled tree sprouts but also increase in secondary compounds that may inhibit herbivory. So is it a trade off? Do the dietary advantages of C and N outweigh the negative effects of the Salicortin compounds? Also, do beavers actually take the juvenile shoots of felled trees or do they avoid them as Basey et al. demonstrated? The leaching aspect you cite in your manuscript was an experimental approach conducted with red maple branches and may not be completely applicable to how you use it as support. Additionally, it has not been repeated. My experience is that beavers do cut and eat red maple but it depends on what other tree species are available. I'd go back and re-read it and see if you still think it applies.
The section on conservation and management is interesting but a bit problematic. I'd move or add a citation after the long quote by the one rancher (I like it and it is a bit different from what my rancher cousin in California/Nevada thinks about beavers and water, but it is anecdotal) and needs a clear citation. The part about beaver activity encouraging invasive species I believe misses the point. Beavers did not put the invasive trees there and are only doing what beavers do as choosy generalists - they choose the species they gain the most from and are most abundant. The way this is written it implies that one way to slow the advance of invasive trees is to restrict beaver activity. This seems extreme since, as you write, they are foundation species and have multiple positive impacts on riparian systems. Just a thought. To me it is much like blaming beavers for moving north as permafrost thaws and changing the subarctic and arctic riparian systems (and maybe increasing climate change).
6. Other thoughts: Your manuscript presents interesting information on chemistry changes in cottonwoods and arthropod abundance that may be due to beaver herbivory. There are a number of issues in regards to timing of data collection and variation between sites. These are impossible to correct due to when the data were collected. I would like to know more about beavers and eastern cottonwoods since you indicate they do cut them but the species does not seems to show the same chemical changes as the other species. You do indicate that this encourages future study, but 12 years have already passed and I wonder if anyone has started to look at this? Also, the lit cited section has lots of capitalization problems.
In summary, with corrections (I'd suggest a careful editing of the entire manuscript), the data you present on chemistry and arthropod abundance on cut vs. uncut cottonwoods adds to our knowledge of how beavers influence riparian systems.
Reviewer 2 Report
I liked the article very much, the description of beavers-trees interactions, the approach to the other trophic levels and its relevance in climate change mitigation.
The work is very well organized, very understandable, and well supported in bibliography. I appreciate the statistical methods used which confer seriousness to the article.
To improve the presented paper, I propose:
- Modify Fig. 2:
a1) I would prefer a normal graphic instead of the presented one because, from my point of view, is easy to see and interpret.
a2) It will be interesting signaled the significant differences with small letters (a and b) to give, visually the greatest information available.
a3) Is very important to put a legend in each axis.
- Insert a new Table:
b1) I would appreciate to see a list of the arthropods used in calculations presented in Table 2. I suggest introducing a Table with this information.